# The Effect of Particle Concentration on Bed Particle Diffusion in Dilute Flows

Patricio A. Moreno-Casas [1,*] , Juan Pablo Toro [2], Sebastián Sepúlveda [1], José Antonio Abell [1], Eduardo González [3] and Joongcheol Paik [4]

[1] Facultad de Ingeniería y Ciencias Aplicadas, Universidad de los Andes, Monseñor Álvaro del Portillo 12455, Las Condes, Santiago 7550000, Chile
[2] Carrera de Ingeniería Civil, Universidad Andres Bello, Antonio Varas 880, Piso 5, Santiago 7500973, Chile
[3] Departamento de Obras Civiles, Universidad Técnica Federico Santa María, Santiago 8940897, Chile
[4] Department of Civil Engineering, Gangneung-Wonju National University, Gangneung 25457, Korea
[*] Correspondence: patriciomoreno@miuandes.cl

**Abstract:** In this paper, we present the simulation results of a Lagrangian particle tracking model that computes the motion of saltating sediment particles, which is considered the most important mode of bedload transport in rivers and channels. The model is one-way coupled to a validated turbulent LES-WALE (Large Eddy Simulation – Wall-Adapting Local Eddy-viscosity) channel flow, i.e., the particles do not affect the computation of the flow velocities and pressures, as suggested for dilute flows. The model addresses the particle trajectories, the collision of the particles with the bottom wall, and collision among particles. The focus of this work is placed on the effect of different particle concentrations and flow intensities (different flow shear stresses) on jump statistics and particle diffusion. Numerical results are validated with experimental laboratory data obtained from the literature for particle diameters in the range of sands. The present results indicate that, at particle concentrations up to 2%, the diffusion coefficients in the streamwise and spanwise directions, $\gamma_x$ and $\gamma_z$, for the local range are nearly constants with a value close to one, corresponding to the ballistic regime. At a concentration of 4%, the largest concentration studied herein, values of $\gamma_x$ and $\gamma_z$ for the local range are slightly smaller, with a representative value of 0.9 regardless of flow intensities. For the intermediate regime, it was found that, on average, $\gamma_x \sim 1.2\gamma_z$ with $\gamma_x$ ranging from 0.6 to 0.85 and $\gamma_z$ within the range 0.45–0.70. For a fixed flow intensity, both diffusion coefficients increase with the particle concentration, which is an indication of the contribution of the collision among particles to particle diffusion. For highly controlled simulation conditions, the differences in particle velocity at a given concentration may change drastically, which should translate to important fluctuations in the computation of sediment transport rates. Finally, the employed computational resources are described as a function of particle concentration. Although the number of total collisions increases linearly with the number of particles, the number of collisions per particle reaches a plateau, thus indicating that there exists an upper limiting value for the number of collisions per particle.

**Keywords:** sediment; bedload transport; particle diffusion; particle concentration; saltation; Eulerian–Lagrangian model

## 1. Introduction

Many natural systems and industrial activities rely on the proper quantification of the transport rates of solid particles [1]. Riverbed erosion and the transport of slurries in the mining industry are examples of processes for which estimations of sediment fluxes are crucial [2,3].

Sediment transport is generally classified as suspended load and bedload [4,5]. In bedload transport, particles are transported either by sliding/rolling or jumping over the bed surface. This last transport mechanism is called saltation and, under natural flow

conditions, is one of the driving forms of bedload motion [4–8]. In most cases, bedload transport is either associated with sand or gravel particles [5]. This paper addresses the mode of particle saltation, which, despite occurring in a layer whose height is approximately two to four particle diameters, plays a key role in river morphology and aquatic habitat [1,5].

To estimate bedload transport rates in gravel and sand bed rivers, many expressions have been developed since the early work of Meyer-Peter–Müller [9]. These expressions typically employ the Shields criterion to compute a dimensionless bedload transport rate $q^* = f(\tau_*, \tau_{*c}, R)$ as a function of the Shields parameter $\tau_*$, the critical Shields parameter $\tau_{*c}$, and the submerged specific gravity $R$. Due to their simplicity, these types of formulations applied to quantify bedload sediment transport are usually preferred over the detailed descriptions of Lagrangian models. However, the estimation of bedload transport rates by semi-empirical formulas based on the Shields criterion may lead to differences between one and two orders of magnitude from laboratory and field observations [10–12]. Therefore, further understanding on the dynamic interaction among sediment particles, turbulent flow, and the bed will help to improve the prediction capacity of bedload transport rates [12–14].

In past decades, researchers have focused on understanding bedload mechanics mainly through experimental observations [15]. The assessment of all the different processes underlying bedload transport has proven quite challenging with experimental techniques [16–19]. This challenge is countered by an increased computational power that is of great help for understanding the interplay between particle motion and near bed turbulent structures [15]. To that end, different numerical approaches have been used in the study of bedload transport, such as Eulerian–Eulerian and Eulerian–Lagrangian methods [20]. The former treats both fluid and particles as continuous phases (Eulerian), whereas the latter treats sediment particles as a dispersed phase (Lagrangian) [21,22]. The Eulerian–Eulerian approach requires appropriate closure laws to describe the physical processes related to the fluid–particle and interparticle interactions [23]. The Eulerian–Eulerian description has been primarily applied to dense bedload transport (i.e., sheet flows) with three main frameworks to close the system [20]: the Bagnold formula [24], the μ(I)-rheology [25,26], and the kinetic theory [27,28]. The Eulerian–Lagrangian approach, also known as computational fluid dynamics–discrete element method (CFD-DEM), provides better results in terms of resolution of the particle–fluid interaction and particle–particle interaction, and no closure laws are required [23,29]. Eulerian–Lagrangian models can use resolved and unresolved methodologies. The resolved methods employ direct numerical simulation (DNS) for solving the flow, whereas surface stresses around each particle are solved by using either arbitrary Lagrangian–Eulerian (ALE) or immersed boundary method (IBM) techniques [29–31]. Although the level of detail achieved by this approach is desired, it is computationally very expensive and limited to small computational domains, small number of particles, and, up to now, unsuitable for large Reynolds numbers. The unresolved methods, instead, use empirical formulas to describe the change of linear and angular momentum for each particle, coupled with formulas that treat the wall–particle and interparticle interactions. The particles are treated as points to which mass, forces, and torques are assigned, which is why this methodology is called the point–particle approach. The point–particle approach has the advantage of explicitly predicting the fluid–particle, particle–particle, and wall–particle interactions with good accuracy, which is very important for studying bedload transport, significantly reducing the computational costs and limitations of the resolved approach [23,29,32].

In this paper, we present numerical results obtained by our in-house C++ code for the Lagrangian calculations of particle saltation, focusing on the combined effects of particle concentration and flow intensity on particle jump statistics and particle diffusion. Given the moderate Reynolds number of the flow ($10^4$), a large eddy simulation (LES) model was considered for the turbulent channel. Several particle concentrations and flow intensities were considered that cover a range of values for which experimental measurements were

available. Details of the computational resources employed for various concentrations are also presented and discussed.

## 2. Description of the Lagrangian Model

### 2.1. Description of the Forces Included in the Saltation Model

According to Newton's second law, a particle of mass $m$ that moves at velocity $\boldsymbol{u_p}$ in a flow of velocity $\boldsymbol{u_f}$ has an acceleration determined by the addition of several forces acting on it. The employed forces selected for this model are submerged weight ($\boldsymbol{F_{sw}}$), drag ($\boldsymbol{F_{dr}}$), lift ($\boldsymbol{F_{lf}}$), Basset ($\boldsymbol{F_{bs}}$), Magnus ($\boldsymbol{F_{mg}}$), added mass ($\boldsymbol{F_{am}}$), and fluid acceleration ($\boldsymbol{F_{fa}}$). Thus, the equation of motion for saltating particles can be written as:

$$m\frac{d\boldsymbol{u_p}}{dt} = \boldsymbol{F_{sw}} + \boldsymbol{F_{dr}} + \boldsymbol{F_{lf}} + \boldsymbol{F_{bs}} + \boldsymbol{F_{mg}} + \boldsymbol{F_{am}} + \boldsymbol{F_{fa}} \tag{1}$$

Most of the terms in Equation (1) for calculating forces are taken from [33]. Equation (1) does not include the effect of collisions during the event of saltation. The interchange of momentum due to collision of two particles is included in the model by using other equations. The selected forces are briefly described below.

### 2.1.1. Submerged Weight ($\boldsymbol{F_{sw}}$)

The submerged weight for a spherical particle can be calculated as the difference between the gravitational force and the buoyancy force due to the displaced volume.

$$\boldsymbol{F_{sw}} = \frac{\pi}{6}\left(\rho_p - \rho\right)g{d_p}^3 \tag{2}$$

where $g$ is the acceleration of gravity, $\rho$ and $\rho_p$ are the water and particle densities, respectively; and $d_p$ is the particle diameter.

### 2.1.2. Drag Force ($\boldsymbol{F_{dr}}$)

This force comes from the effect of pressure and viscous forces acting on the surface of the particle in the direction of the relative flow velocity. The drag force, expressed in terms of the relative particle velocity in the flow direction $\boldsymbol{u_r}$, can be written as:

$$\boldsymbol{F_{dr}} = -\frac{3}{4}C_D\frac{\rho}{\rho_p}\frac{A|\boldsymbol{u_r}|U_r}{d_p} \tag{3}$$

where $C_D$ is the drag coefficient, $A$ is the particle cross section in the direction of $\boldsymbol{u_r}$, and $U_r$ is the magnitude of $\boldsymbol{u_r}$. The relative velocity of the particle can be estimated as $\boldsymbol{u_r} = \boldsymbol{u_p} - \boldsymbol{u_f}$.

### 2.1.3. Lift Force ($\boldsymbol{F_{lf}}$)

For spherical particles under the presence of shear stresses, a pressure gradient normal to shear stresses will be generated, which produces lift. This force can be written as:

$$\boldsymbol{F_{lf}} = \frac{3}{4}\rho\frac{C_L}{d_p}\left(|\boldsymbol{u_r}|_T^2 - |\boldsymbol{u_r}|_B^2\right) \tag{4}$$

where $C_L$ is the lift coefficient, and $|\boldsymbol{u_r}|_T$ and $|u_r|_B$ represent the relative velocity at the top and bottom regions of the particle, respectively.

### 2.1.4. Basset Force ($\boldsymbol{F_{bs}}$)

The Basset force or history force accounts for the delay in boundary layer development due to the instability of the flow around an accelerating particle, and it can be computed as:

$$F_{bs} = -\frac{9}{d_p}\left(\frac{\rho\mu}{\pi}\right)^{0.5}\int_0^t \frac{d\boldsymbol{u_r}}{d\tau}\frac{d\tau}{(t-\tau)^{0.5}} \tag{5}$$

where $t$ is time, $\mu$ is the dynamic fluid viscosity, and $\tau$ is the integration variable. The above equation must be solved carefully because the upper integration limit is singular (see, for example, [34]).

2.1.5. Magnus Force ($F_{mg}$)

The particle rotation under the effect of a flow field generates a force that is perpendicular to the direction of the particle motion and rotation. It is also called rotation lift force. A basic form of Magnus force for high Reynolds numbers corresponds to:

$$F_{mg} = C_M \pi \rho \frac{d_p^3}{8}\left(\boldsymbol{\Omega_p} \times \boldsymbol{u_r}\right) \tag{6}$$

where $\boldsymbol{\Omega_p}$ represents the angular velocity of the particle, and $C_M$ (ranging from 0.4 to 0.55) is the Magnus coefficient that relates the Magnus force with the Reynolds number, and rotation with the particle relative velocity.

2.1.6. Added Mass Force ($F_{am}$)

The particle motion in a fluid necessarily requires the displacement of the fluid surrounding the particle. This effect is quantified through the added mass force or virtual mass. For a spherical particle immersed in a high Reynolds number flow, this force can be written as follows:

$$F_{am} = \rho C_m \frac{d}{dt}\left(\boldsymbol{u_f} - \boldsymbol{u_p}\right) \tag{7}$$

where $C_m = 0.5$ is the added mass coefficient.

2.1.7. Fluid Acceleration Force ($F_{fa}$)

This force is attributed to the fluid motion away from the particle when the control volume analysis is applied. It can be written as:

$$F_{fa} = \rho \frac{D\boldsymbol{u_f}}{Dt} \tag{8}$$

*2.2. Sub-Model for the Particle Free-Flight*

The 3D sub-model proposed herein describes the trajectory and velocity of free flying particles by describing the hydrodynamic forces responsible for the particle motion. The equations are combined with an angular momentum equation devoted to follow in time the particle rotation.

The equations of the sub-model provide the non-dimensional particle velocities in the streamwise, wall normal, and spanwise directions, respectively, as it can be seen in Figure 1 ($u_p$, $v_p$, and $w_p$). In addition, each particle has an angular velocity, and it can collide with the bed or other particles. The channel bed has an inclination with respect to the horizontal plane ($\theta$), where particles can move and collide with the wall (formed with packed particles of the same diameter) and other moving particles, as shown in the following figure.

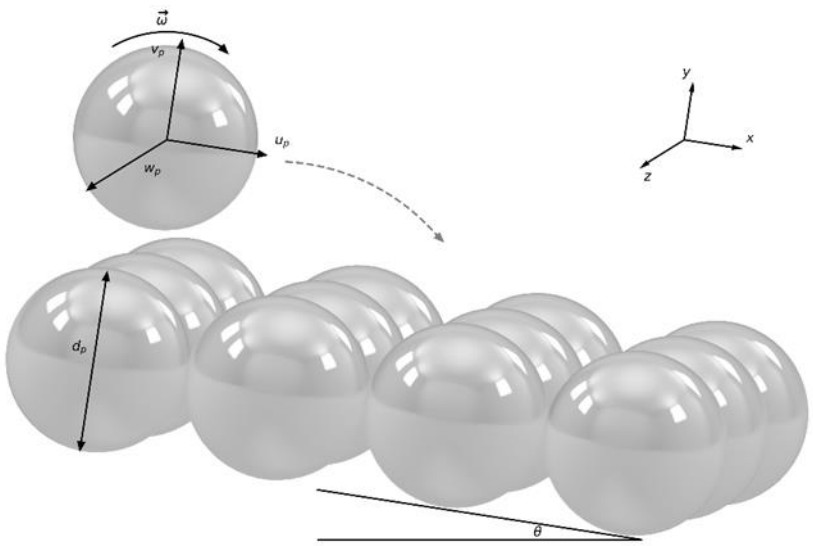

**Figure 1.** Schematic of the 3D particle jumps.

The momentum equations presented below are written in non-dimensional form and include the particle diameter as length scale and the shear velocity $u_*$ as a velocity scale.

$$\frac{du_p}{dt} = \alpha \frac{\sin \theta}{\tau_*} - \frac{3}{4}\alpha C_D \left(u_p - u_f\right)|\boldsymbol{u_r}| + \alpha C_m w_p \frac{du_f}{dz} + \frac{9\alpha}{\sqrt{\pi R_p \tau_*^{\frac{1}{4}}}} \int_0^t \frac{d}{d\tau}\left(u_f - u_p\right)\frac{d\tau}{\sqrt{t-\tau}} + \frac{\alpha Du_f}{Dt} \tag{9}$$

$$\frac{dv_p}{dt} = -\frac{3}{4}\alpha C_D \left(v_p - v_f\right)|\boldsymbol{u_r}| + \alpha C_m \frac{dv_f}{dt} + \frac{9\alpha}{\sqrt{\pi R_p \tau_*^{\frac{1}{4}}}} \int_0^t \frac{d}{d\tau}\left(v_f - v_p\right)\frac{d\tau}{\sqrt{t-\tau}} + \frac{\alpha Dv_f}{Dt} \tag{10}$$

$$\begin{aligned}\frac{dw_p}{dt} = -\alpha \frac{\cos\theta}{\tau_*} &- \frac{3}{4}\alpha C_D \left(w_p - w_f\right)|\boldsymbol{u_r}| + \alpha C_m \frac{dw_f}{dt} + \frac{9\alpha}{\sqrt{\pi R_p \tau_*^{\frac{1}{4}}}}\int_0^t \frac{d}{d\tau}\left(w_f - w_p\right)\frac{d\tau}{\sqrt{t-\tau}} + \frac{\alpha Dw_f}{Dt} \\ &+ \frac{3}{4}\alpha C_L \left({|\boldsymbol{u_r}|^2}_T - {|\boldsymbol{u_r}|^2}_B\right) + \frac{3}{4}\alpha|u_r|\left(\varpi_y - \frac{1}{2}\frac{du_f}{dz}\right)\end{aligned} \tag{11}$$

In Equations (9)–(11), $u_f$ is the streamwise velocity component of the fluid, $\alpha$ is defined as $(1 + R + C_m)^{-1}$, and $R$ is the submerged specific gravity of the particle, which is defined as $(\rho_s/\rho - 1)$; $\tau_*$ is calculated as $u_*^2/(gRd_p)$; $g$ is gravity; $R_p = \left(gRd_p^3\right)^{0.5}/\nu$ is the explicit Reynolds number of the particle; $\nu$ is the kinematic viscosity of water; $\theta$ is the angle of the bed with respect to the horizontal plane; $\varpi_y = \omega_y d_p/u_*$ denotes the non-dimensional component of the rotation vector in the spanwise direction; $\omega_y$ represents the angular rotation of the particle along the spanwise direction; and $y$ corresponds to the direction normal to the bottom of the channel.

The terms on the right-hand side of the Equation (9) correspond to the following forces per unit mass: submerged weight or buoyant force; non-linear drag force; the reminder of the added mass or virtual mass force; Basset force, and fluid acceleration force, respectively. In Equation (10), the right-hand side terms are non-linear drag force, the reminder of the added mass or virtual force, Basset force, and fluid acceleration force, respectively. Finally, in Equation (11), the terms represent the submerged weight or buoyant force, the non-linear drag force, the reminder of the added force, Basset force, fluid acceleration force, the lift force, and the Magnus force, respectively. The operator $d(\cdot)/dt$ indicates the material derivative using the particle velocity. A value of $C_m = 0.5$ for the coefficient of virtual mass was employed as in other particle tracking models. As suggested by [35], the selected lift coefficient was $C_L = 0.2$.

The drag coefficient, $C_D$, was calculated using the expression proposed by [36]:

$$C_D = \frac{24}{Re_p}\left(1 + 0.15\sqrt{Re_p} + 0.017Re_p\right) - \frac{0.208}{1 + 10^4 Re_p^{-0.5}} \qquad (12)$$

where $Re_p = v_s d_p / \nu$ is the particle Reynolds number, and $v_s$ is the particle fall velocity. The non-dimensional rotation vector $\varpi$ is numerically calculated at each time step using the expression proposed by [37]:

$$\frac{d\varpi}{dt} = -C_t \frac{15}{16\pi}|\varpi_r|\varpi_r \qquad (13)$$

where $C_t = C_1/\sqrt{Re_r} + C_2/Re_r + C_3 Re_r$; the Reynolds number associated to the rotation motion is a non-dimensional coefficient written as $Re_r = d_p^2|\varpi_r|/4\nu$; the coefficients $C_1$, $C_2$, and $C_3$ are taken from the table presented in [37]; and $\varpi_r$ is the non-dimensional vector of relative rotation of the particle with respect to the fluid vorticity.

Equations (9)–(11) are a 3D extension of the equations presented by [33]. They are based on the point–particle approach [21], and they include all forces explained in Section 2.1 of the present paper (see [4]). Except for the Basset term, Equations (9)–(11), are solved by using the standard fourth order Runge–Kutta method. The Basset force is treated differently because the integral is singular at the upper integration limit. The second order methodology analyzed in [32] has been used to circumvent this problem. The simplifications applied to the above equations are (a) the stream-wise and span-wise components of the lift force are close to zero; and (b) the stream-wise and wall-normal components of the Magnus force are negligible, an assumption that has been corroborated through many computations (see [4]).

*2.3. Sub-Model for Collision of Particles with the Bottom Wall*

The numerical simulation for the process of particles colliding with the bottom wall is usually separated into three sub-models that represent the following features: the rebound of the particle and the representation of the roughness of the bottom wall.

The rebound sub-model allows one to calculate the linear and angular velocities of the particle after the collision based on the linear and angular velocity just before the collision with the wall. In addition, the energy loss during the semi-elastic collision between particles and wall is considered by using the friction and restitution coefficients. The irregularities of the bottom wall inherent to any streambed are modeled through expressions that include geometric and random terms, allowing the variability of jump lengths and jump heights, as well as the global diffusion of the particles.

The 3D model employed in this work was proposed by [38]. The model includes the conservation of linear and angular momentum before and after the collision with the bottom wall. The post-collision velocity, expressed as a function of the particle velocity immediately before the collision, is calculated depending on whether the particle slides over the bottom wall. The equation was developed for the contact with the horizontal plane among the particle and the bottom wall [22]. An advantage of this model is that it can be easily extended to simulate the collision among particles.

*2.4. Sub-Model for Collision among Particles*

When two or more particles in very close proximity are about to collide, then a subroutine for calculating the linear and angular velocities after the collision is used. For collision among particles, two classes of models are generally considered: the hard-sphere model and the soft-sphere model [22]. The first model includes collision between only two particles (binary collision), which is a good approximation for flows with low particle concentrations. This model includes the use of restitution, $e$, and friction, $f$, coefficients to calculate the interchange of linear and angular momentum. In contrast, the soft-sphere approach employs elements from mechanics to simulate collisions among multiple particles

and to estimate the post-collision linear and angular velocities. This feature makes the model very useful when the particle concentration is high, given that, in this case, multiple particles may collide at the same time. However, the computational cost is larger than that needed for the hard-sphere model. Given that particle concentrations as high as 5% are considered in this work, the soft-sphere model will be applied. For more information regarding both collision models, please refer to [22].

### 2.5. Diffusion

A particle moving as bedload can be at rest or in motion (either rolling/sliding or saltating). A particle moving in saltation mode moves due to the hydrodynamic forces acting on it. At some point, this particle may be trapped during the collision with the bed. Another reason for a particle to remain in the bed is simply due to a significant decrease in the flow intensity. In this case, the flow is not strong enough to set the particle in motion. The particle can then be in suspension again due to an increase in the net hydrodynamic force acting on it locally. Therefore, in the long-term trajectory of a particle in saltation mode, there are many collisions with the bed, periods of rest, and resuspensions to consider. If a particle transported by the flow substantially changes its position in time, there is said to be great diffusion in a certain axis (X and Z-axes in our case). If a particle follows a nearly straight line in the X-axis, then diffusion in the Z-axis is zero. The authors in [39] suggest a conceptual model to estimate diffusion of particles for bedload transport, identifying three ranges of spatial and temporal scales with different diffusion regimes, namely: local range, intermediate range, and global range. By diffusion, it is meant the rate in time at which particles disperse in plan view (X-Z plane). No diffusion is considered in the Y-axis.

The local range corresponds to ballistic trajectories of particles, which occur between two successive collisions with the bottom wall ($\gamma_x \approx \gamma_z \approx 1$). These trajectories are the result of the inertial motion of the particle, with no rests or abrupt changes in particle position. The intermediate range corresponds to trajectories of particles between two successive rests or periods of rest. Trajectories in this range consist of many local trajectories, including dozens or hundreds of collisions with the channel bed. In the intermediate range, diffusion can, in principle, be slow/sub-diffusive ($\gamma_x, \gamma_z < 0.5$), normal/Gaussian ($\gamma_x = \gamma_z = 0.5$), fast/superdiffusive ($\gamma_x, \gamma_z > 0.5$), or a mixture of values depending on factors driving the transport process. For example, the bed bathymetry and turbulence close to the bed may have opposite effects on the diffusion of particles in saltating mode. The bed roughness can slow the process of diffusion, whereas turbulence can enhance it. The global range of scales corresponds to particle trajectories that consist of many intermediate trajectories, as intermediate trajectories consist of many local trajectories. The behavior of the particle in the global range of scales is most probably sub-diffusive ($\gamma < 0.5$), because of multiple periods of rest. The local, intermediate, and global ranges are taken from the conceptual model of [39], and the curves in that model are given by the following regression equations:

$$\langle X'^2 \rangle = \alpha_x t^{2\gamma_x} \tag{14}$$

$$\langle Z'^2 \rangle = \alpha_z t^{2\gamma_z} \tag{15}$$

where $X'^2$ and $Z'^2$ are the second order moment of the particle position in X and Z, respectively; $\alpha_x$, and $\alpha_z$ are constants; $t$ is time; $X' = X - \overline{X}$, $Z' = Z - \overline{Z}$, with $\overline{X}$ and $\overline{Z}$ being mean values. The exponents $\gamma_x$ and $\gamma_z$ denote the state of diffusion, and $\langle \, \rangle$ indicate ensemble average.

According to the experimental values found by [39,40], it is expected that diffusion in the spanwise direction (Z) should be approximately ballistic in the local range and super-diffusive in the intermediate range. In this work, the global range is not modeled because the saltation code does not include particle rests. The conceptual model of [39], together with the average values of jump height and jump length, allow the validation of the tridimensional model.

## 3. Study Case

The experimental data, the main features of the computational simulations, the description of the Lagrangian model, and a brief description of the analyzed cases are presented below.

### 3.1. Experimental Information for Validation

To validate the saltation model, it is necessary to check that our numerical results are coherent with the experimental results available in the literature. Experimental information for sediment transport with diameters in the range of sands can be found in [8,41–43]. As presented in Table 1, those articles contain detailed information on trajectories of saltating particles, including particle diameter, number of jumps, shear velocity as a measure of the flow intensity, and statistics of jumps; *H* and *L* represent non-dimensional values for mean particle height and length, respectively, obtained from experiments.

**Table 1.** Characteristics of experimental information for particle jumps.

| Authors | Recording Method | Particle Size (mm) | Number of Jumps | $u_*$ (m/s) | Statistics for $H$, $L$, and $u_p$ |
|---|---|---|---|---|---|
| Lee et al. (2006) [41] | Standard video camera | 0.6 | Not available | 0.039–0.068 | Mean values |
| Lee et al. (2000) [44] | Standard video camera | 6 | Not available | 0.038–0.054 | Mean values |
| Niño and García (1998) [42] | High-speed video camera | 0.5–0.8 | 1–2 jumps every 100 particles | 0.021–0.026 | Mean and standard deviation |
| Niño et al. (1994) [43] | Standard video camera | 15–31 | 80 | 0.14–0.23 | Mean and standard deviation |
| Lee and Hsu (1994) [8] | Standard video camera | 1.36–2.47 | Not available | 0.036–0.105 | Mean values |

The experiments were carried out by using sands $\left(d_p = 0.0625 - 2 \; mm\right)$ and gravels $\left(d_p = 2 - 64 \; mm\right)$, which correspond to the most common type of non-cohesive sediments transported in natural streams via saltation.

In Table 2, the recommended values for the friction and restitution coefficients according to three authors are presented. These coefficients are employed in the collision sub-models, and their values strongly depend on the material of the saltating particle and the flow conditions, thus no universal values are available. In this work, the values suggested by [45] are employed, based on [4], who analyzed a large dataset of particle jump heights for two sizes of sands, confirming the use of the values given by [45].

**Table 2.** Values for friction and restitution coefficients.

| Authors | Restitution Coefficient $e$ | Friction Coefficient $f$ |
|---|---|---|
| Niño and García (1994) [33] | $0.75 - 0.25 \; \tau_* / \tau_{*c}$ | 0.89 |
| Schmeeckle et al. (2001) [45] | 0.65 | 0.10 |
| Tsuji et al. (1987) [46] | 0.80 | 0.40 |

### 3.2. Computational Implementation

The simulation was performed with the open-source software OpenFOAM 5.x, which consists of more than 100 libraries written in C++. The particle code was modified and optimized to run in parallel to substantially increase the number of particles, and it corresponds to an extension from a previous saltation code, written in Fortran 90 [4]. For that purpose, the new code has been written in C++ and parallelized using OpenMP and a cell-based space partitioning algorithm. Space is subdivided into sub-volumes or cells,

which contain the particles, and each cell is assigned to a process to compute the state update of its particles in parallel. In all these processes, for each cell belonging to the process, the particle–particle collision subroutine only verifies particle collisions for those particles that are within the cell or on neighboring cells, which significantly reduces computational complexity of checking collisions on all pairs of particles. The code is extendable to run in two-way coupling mode (with OpenFOAM) by applying a few modifications. The algorithmic speedup thus obtained allows the simulation of dilute particle flows (particle concentrations by volume up to 5%) for orders of O(100000) particles in a conventional server. The study of higher concentrations within the dilute particle flow range and on the non-dilute flow regime (concentrations above 5% by volume) requires the use of particle tracking codes with parallel capabilities, and an efficient use of memory to manage the large amounts of data storage needed at each time-step for high particle concentrations. This work is the first step on the use of this code on the study of higher concentration particle laden flows, which are of interest in industrial and natural flow studies. The modeled particles are in the range of sands with a particle diameter of $d_p = 0.69$ mm. The model was validated with experimental observations given by [42]. The employed friction and restitution coefficients correspond to the ones suggested by [45], which agree with the numerical tests developed by [4].

The Reynolds number of the flow, based on a friction velocity of $u_* = 0.024$ m/s and the channel height $h = 0.028$ m, is $Re_\tau = 590$ corresponding to a moderate Reynolds number ($Re \sim 10^4$). In view of this, an LES model is used for reproducing the turbulent channel flow. The sub-grid eddy viscosity is obtained from the WALE model [47]. Periodic boundary conditions were employed at the inlet, outlet, and at the sides of the computational domain. The size of the computational domain is equal to $2\pi h \times \pi h \times 2h$ in the streamwise, spanwise, and vertical directions, respectively, as shown in Figure 2a. The number of computational cells was equal to $1.32 \times 10^6$ with $120 \times 100 \times 110$ volumes in the streamwise, spanwise, and vertical directions, respectively. The same cell size was employed in the streamwise and spanwise directions, whereas in the vertical direction, normal to the bottom wall, an expansion ratio of 1.20 was included. The first grid cell was located at $\overline{y_1^+} = \overline{u_* y_1}/\nu \sim 0.5$ (where $y_1$ corresponds to the distance from the wall to half of the first finite volume). The computational domain, depicted in Figure 2a, shows a very fine grid distribution near the walls. This distribution is chosen to obtain a good resolution of the velocity gradients where the sediment particles move. The WALE sub-grid model was applied herein as it gives good results near the wall, where sediment transport takes place. Therefore, the mesh is more refined near the wall, until having finite volumes within the viscous sublayer ($y^+$ less than 5) and being able to capture the different scales of turbulence. Once the turbulent flow is solved, the velocity field for each time is stored. The information is then used for running the one-way coupled Lagrangian model for particle saltation. Figure 2b shows the velocity magnitude resolution achieved by the LES-WALE model near the wall. The flow simulation has been setup to mimic the experimental flow conditions achieved in the simulations by Niño and García [42], which will be used to validate the particle jump statistics.

To validate the Lagrangian model, the same experimental conditions presented by [42] were simulated. These experiments provide statistics of mean and standard deviation for three non-dimensional parameters: the particle jump length (*L*), jump height (*H*), and stream-wise velocity ($u_p$). A single particle trajectory was simulated for a value of $R_p = 73$. Three different flow intensities were simulated; corresponding non-dimensional shear stresses of $\tau_*/\tau_{*c} = 1.79$, 2.43, and 2.67 were employed to that end, where $\tau_{*c}$ is the critical shear stress associated with the incipient motion of the particle. For the particle size utilized in this study, $\tau_{*c} = 0.032$.

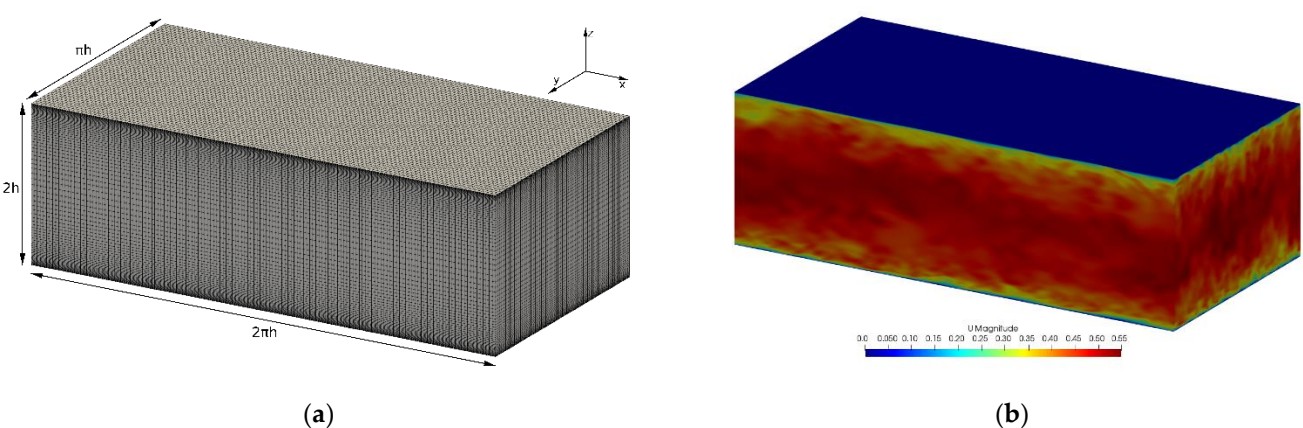

(**a**)                  (**b**)

**Figure 2.** Grid and computational domain employed in the LES simulation using periodical flow. (**a**) Grid distribution and dimensions of the computational domain, (**b**) sample of LES velocity magnitude (m/s) at a given time step.

As an initial condition, the particle has a predefined angular and linear velocity. Particles located at the bottom wall are of the same diameter as the particles in motion. For simplicity, all particles are perfect spheres with the same diameter. Simulation time was 200 s, where an average of 120 jumps were obtained for each case. To remove the effect of the initial conditions, the first 10 jumps were removed from the statistical analysis. Both mean values and standard deviation were calculated with the same methodology. In Figure 3, an example of the particle trajectories (top view), plotted in MATLAB, is presented, where the differences of particle mobility are very clear as particle concentration changes. Diffusion of particles on the transverse direction is also evident, where particles use a wider section of the computational domain as the particle concentration increases.

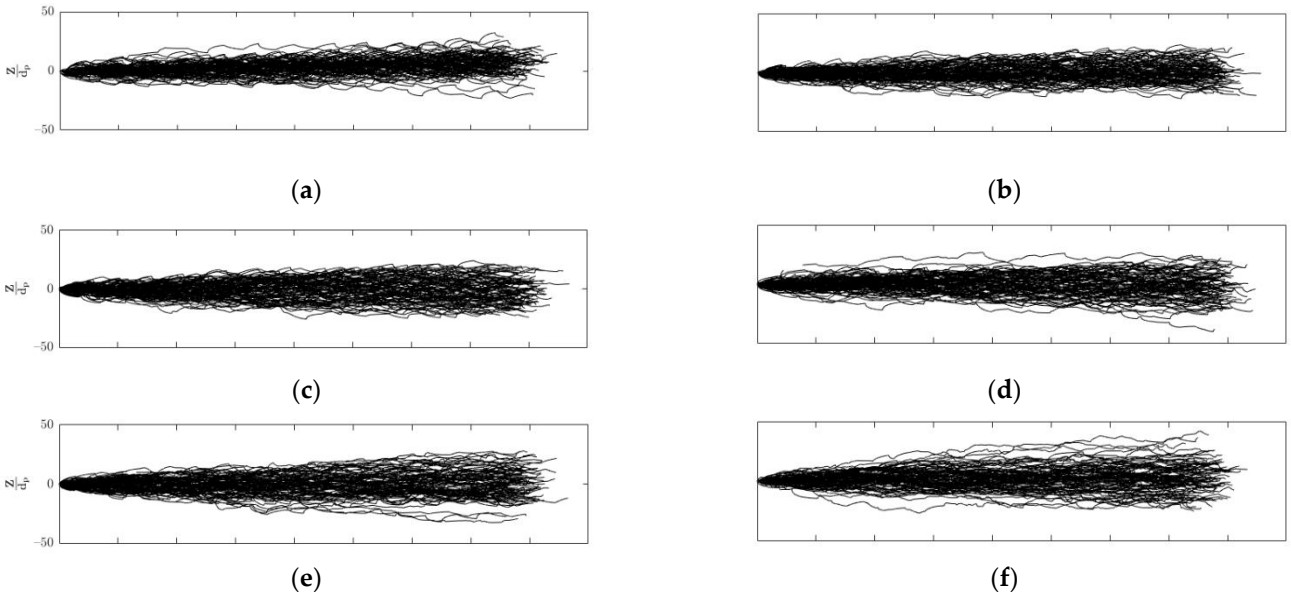

(**a**)                  (**b**)

(**c**)                  (**d**)

(**e**)                  (**f**)

**Figure 3.** *Cont.*

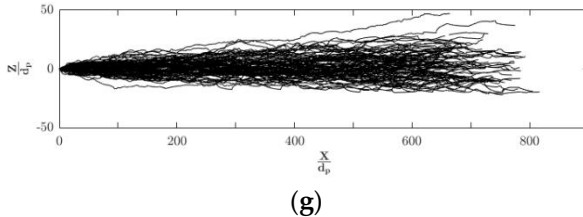

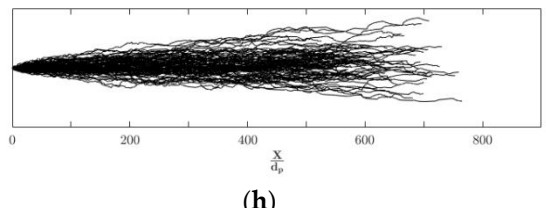

(**g**)  (**h**)

**Figure 3.** Top view (X-Z axes) sample of particles moving with different particle concentrations for $\tau_*/\tau_{*c}$ =1.79: (**a**) 1000 particles; (**b**) 3000 particles; (**c**) 4500 particles; (**d**) 6500 particles; (**e**) 13,000 particles; (**f**) 26,000 particles; (**g**) 52,000 particles; and (**h**) 99,000 particles.

### 3.3. Description of How the Model Works

To begin, the computational domain where the process of saltating particles will take place is defined (see Figure 1). Because the process is one-way coupled, the flow is solved first and then the obtained velocity fields are employed to simulate the particle motion. The Lagrangian code to follow the particles is employed in the same computational domain used for simulating the turbulent flow. By creating the cells, each one will have its own properties, such as flow velocity and vorticity, in the three spatial coordinates. The cell number, cell location and dimensions, and the number of particles inside each cell are also defined. The following step is to create the particles. To that end, a cell (near the wall) is chosen randomly and a particle is created inside that cell. Each particle will have properties, such as velocity and rotation, where assigned initial values have been generated randomly within the range of the experimental values of [39]. Likewise, the simulation time, particle number, and particle location are predefined as simulation input. The simulation can then be carried out. The particle algorithm goes through each cell until a particle is found. Once all particles have been identified, their new velocity is computed and time is updated. By doing this, we avoid moving a particle twice during the same time step.

In order to compute the particle new velocity, there are three different options linked to the three sub-models depending on whether the particle hits the bottom wall, the particle hits another particle, or the particle moves in a free trajectory. In this last case, the hydrodynamic forces are responsible for modifying the particle velocity. This sequence is repeated until the simulation time is completed. During these processes, important information is stored as text files. This includes jump height, jump length, and number of jumps. Values of position, velocity, and rotation of the particles are also stored. Once the simulation is finished, the relevant statistics are calculated. It is noteworthy that the model was built to run in parallel with more than one processor, so the calculation time is reduced, allowing for the computation of thousands of particles.

The input values defined for the model before each run are: number of particles, total simulation time, particle id, number of processors, to activate/deactivate selected forces, to activate/deactivate collisions among particles, source of flow velocity field (either logarithmic velocity profile or simulated turbulent flow (OpenFOAM)), and, finally, the corresponding flow shear stress ($\tau_*$).

There are two types of simulation results provided by the model: mean and instantaneous values. Mean values include average jump length and jump height, with the corresponding standard deviation values. Instantaneous values include the position and the linear and angular velocity for each particle. Fluid velocity and vorticity values for each cell where a particle is located are also stored. Finally, the number of collisions among particles for each particle is also obtained.

### 3.4. Cases to Be Analyzed

The effect of increasing particle concentration per unit volume was analyzed. Simulations using eight different particle concentrations (by volume) were carried out: 0.04% (1000 particles), 0.12% (3000 particles), 0.18% (4500 particles), 0.26% (6500 particles), 0.52% (13,000 particles), 1.03% (26,000 particles), 2.07% (52,000 particles), and 3.94% (99,000 par-

ticles), which are concentrations in the dilute range (up to 5% per volume). Various flow shear stresses were also included to validate the model.

To estimate particle concentration, the computational domain used one half of the distance between the parallel plates. The effects of increasing particle concentration on the jump statistics, motion of particles, and computational resources spent in the simulation are analyzed in what follows.

The statistical characterization of jumps includes the dimensionless mean value of jump height and jump length and their corresponding standard deviations. The parameters of particle motion are the particle velocity and rotation. The computational effort was assessed by comparing the resulting execution time among the two main functions of the model. The first one is the required time to calculate the new positions for each particle, and the second one is the required time to calculate velocities for the next time step. The latter includes collisions among particles and collisions with the bottom wall. In Table 3, the central processing unit (CPU) time spent on simulations is presented.

**Table 3.** CPU time (s) to carry out simulations.

| Number of Particles | Particle Concentration (%) * | CPU Time (s) |
| --- | --- | --- |
| 1000 | 0.04 | 3251 |
| 3000 | 0.12 | 3548 |
| 4500 | 0.18 | 3761 |
| 6500 | 0.26 | 4053 |
| 13,000 | 0.52 | 5088 |
| 26,000 | 1.03 | 7662 |
| 52,000 | 2.07 | 14,543 |
| 99,000 | 3.94 | 31,858 |

Note: * indicates Concentration by volume.

Diffusion analysis is conducted in the X-Z plane, where X, Y, and Z are the streamwise, wall-normal, and spanwise directions.

## 4. Results and Discussion

### 4.1. Model Validation

The numerical results for the 3D saltating motion of particles in a turbulent open channel are presented below. In Figure 4, the statistics of dimensionless jump height, jump length, and stream-wise particle velocity are presented. Numerical values are compared with the experimental results presented in [42]. The range of values for two standard deviations is also presented. Particle statistics were calculated from the information gathered for all simulated particles, where all particles have experienced at least one hundred jumps. From the experimental values presented in [39], jump height and length slightly increase with increasing flow intensity. This trend is also observed in the numerical results (see Figure 4a,b). Figure 4b shows good agreement of particle jump length with the experimental results for flow intensities ($\tau_* / \tau_{*c}$) between 2.43 and 2.67. For moderate flow intensities ($\tau_* / \tau_{*c} = 1.79$ and 1.87), numerical results tend to underpredict when compared with the experimental information.

Simulated mean jump heights and lengths are within the range of one standard deviation observed in the experimental results. It is worth noting that numerical results have better agreement with experiments at higher flow intensities.

As for jump height and length, particle velocities in the streamwise direction increase with increasing flow intensities (see Figure 4c). For moderate flow intensities, numerical and experimental results are in excellent agreement when comparing mean particle velocities, whereas for larger flow intensities, the numerical results slightly overpredict the experimental values. Nevertheless, the numerical values are within the range of one standard deviation.

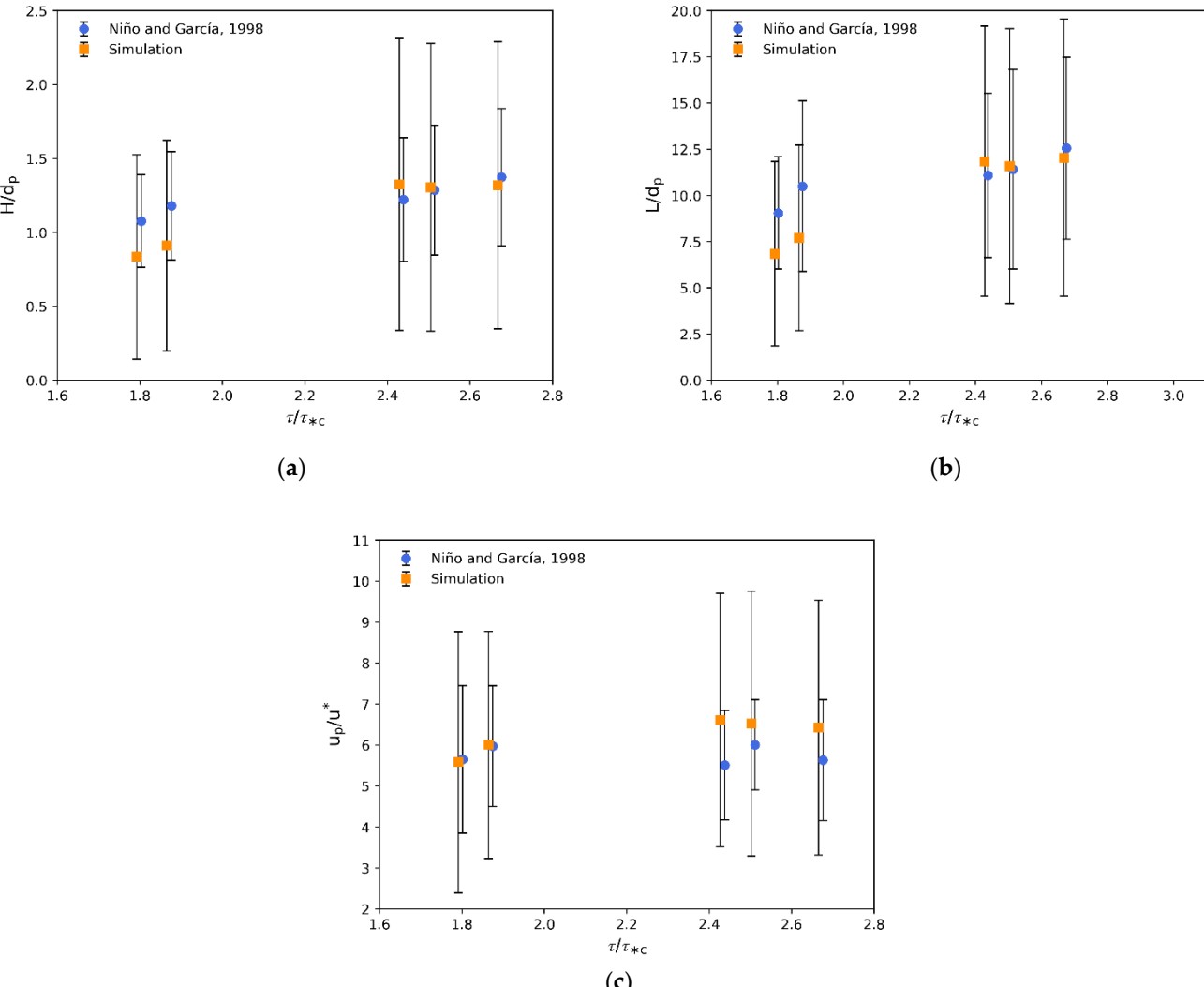

**Figure 4.** Comparison of simulation results with experimental results from Niño and García (1998) [42]. (**a**) Dimensionless average jump height, (**b**) dimensionless average jump length, and (**c**) dimensionless particle streamwise velocity. Vertical lines indicate two standard deviations.

*4.2. Particle Concentration*

As the number of transported particles increases, the likelihood of collision among them also increases. During this process, particles loose energy and change their direction of motion. As concentration increases, particle trajectories are hampered by the action of other particles. In Figure 5, results of jump height, jump length, particle velocity, and rotation are presented. Only three different concentrations per volume are shown, for the sake of clarity, where the effects of increasing concentration on jump statistics, even at small concentrations, can be clearly observed.

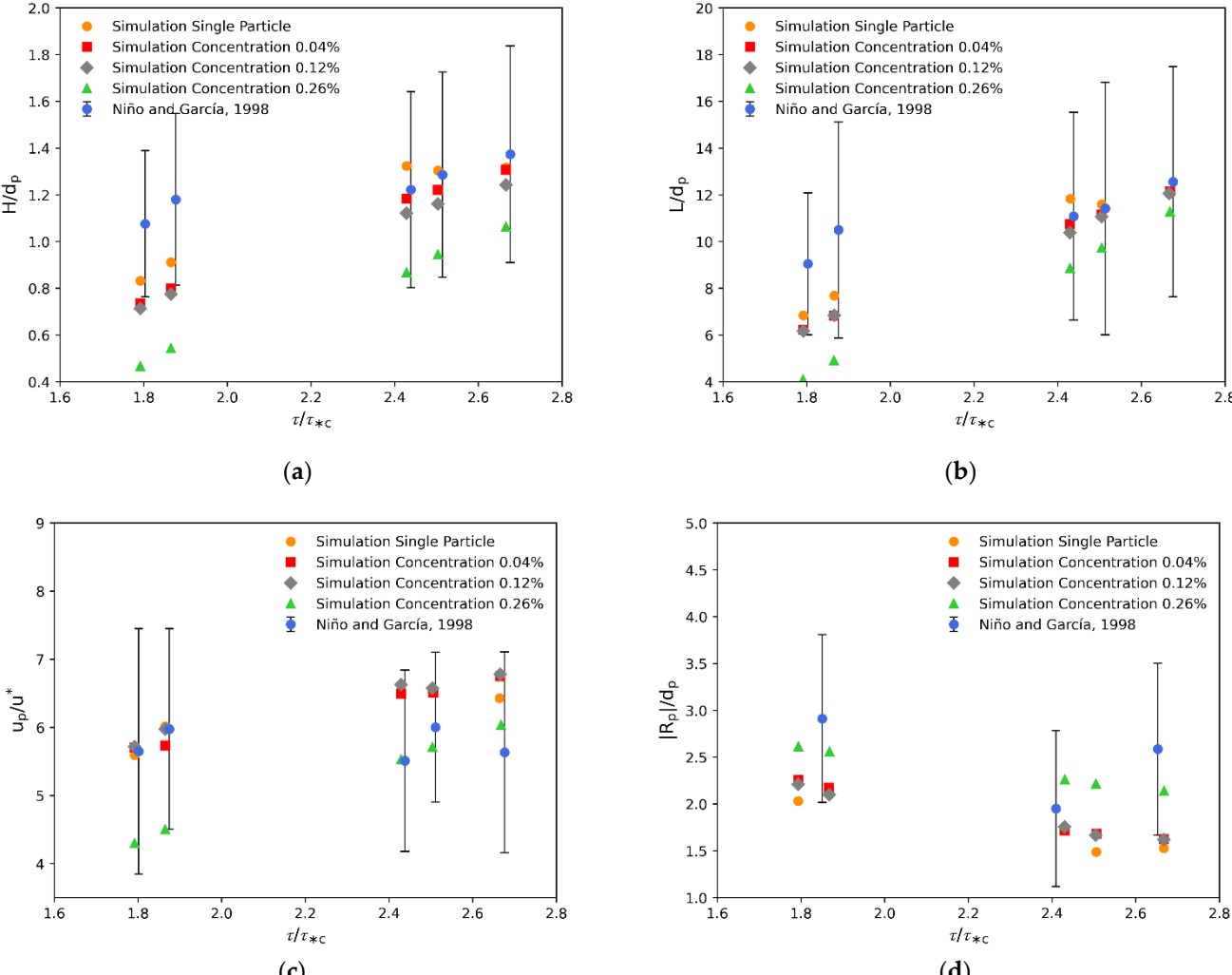

**Figure 5.** Sample of multiparticle simulation results for concentrations of 0.04, 0.12, and 0.26% by volume, compared against experimental results from Niño and García (1998) [42] and simulation from a single particle. (**a**) Dimensionless average jump height, (**b**) dimensionless average jump length, (**c**) dimensionless particle streamwise velocity, and (**d**) dimensionless particle rotation in the spanwise direction. Only standard deviations from Niño and García are shown for the sake of clarity.

In Figure 5a, particle jump height values seem to be less affected by particle concentration as flow intensities increase. It can also be noticed that for the same flow intensity (shear stress), larger particle concentrations generate smaller mean particle jump heights. This is explained by the fact that as particle concentration increases, inter particle collisions also increase, limiting particle motion and therefore reducing the jump height reached by particles.

The effects of particle concentration for different particle intensities on mean particle jump length is illustrated in Figure 5b. An increase in particle concentration results in smaller jump lengths for a fixed flow intensity; however, the effect is reduced as flow shear stresses rise. It may be concluded that at large flow intensities, the momentum exerted by the flow to the particles become more important than the momentum exchange caused by collision among particles, counteracting its effect in terms of mean particle jump length.

The numerical results for the mean particle velocity in the streamwise direction are presented in Figure 5c. At lower particle concentrations at a given flow intensity, it is observed that the effect on particle velocity is very limited. However, for a concentration of 0.26%, there is a clear decrease in mean particle velocity for all flow intensities. This is an indication that a certain threshold for particle concentration should be exceeded to influence velocities and this threshold is close to 0.26%, which is still a very low concentration.

Finally, the magnitude of the particle rotation with respect to the Y-axis is presented in Figure 5d. It is observed that particle rotation decreases with increasing flow shear stresses, showing an inverse proportionality. This is true for all concentrations. At particle concentrations of 0.26% at any given flow intensity, the particle rotation is clearly increased. In other words, at low particle concentrations and a fixed flow intensity, there is no clear trend for the values of particle rotation, except for the 0.26% particle concentration.

Figure 6a shows that a change in particle mean velocity ($\pm$ one standard deviation) is significant as the concentration of particles increases. From concentrations of 0.52%, a change in the mean velocity can be observed. Furthermore, the variation of particle velocities becomes much larger as particle concentration increases. This has a very important effect on the bedload transport, as different particle velocities may cause fluctuations on the rates of bedload transport. This kind of effect is not captured by Meyer-Peter–Müller [9] sediment-transport-type equations. Figure 6b shows boxplots of the dimensionless particle velocity statistics, where median values (red lines) and the 25th and 75th percentiles (top and bottom lines of the blue boxes) are contrasted with the extreme values (vertical dashed lines) obtained by the simulations as the particle concentration changes. The beauty of the boxplot is that it shows extreme values that are not captured by standard deviations (Figure 6a). Figure 6b shows that dimensionless particle velocities, for the same particle concentration, for some particles may be as low as less than one dimensionless unit, and for others as large as nine dimensionless units. This is in line with the findings of many authors (for example, see reviews by [10,13]), where even at very controlled flume experiments, bedload transport rates show large fluctuations, causing great difficulties in the prediction of transport rates in uncontrolled, more realistic conditions, such us rivers and streams. According to the results presented herein, particle velocity fluctuations may be one of the reasons for these fluctuations, all the time that the bedload transport rates (this is how many particles move through a predefined area) are directly correlated to the particle velocity. As sediment transport rate estimations are commonly computed considering the mean particle statistics, thus not including the particle velocity fluctuations, the estimations may differ from the real statistics in several orders of magnitude, even for strictly controlled flow conditions. Simulations with larger particle domains, with more realistic stream beds, and with a realistic sediment granulometry will most likely increase these fluctuations. The use of particle algorithms such as the one presented in this study may help with understanding those variables causing the fluctuations, and contribute to putting forward new expressions to estimate bedload transport rates, new expressions that make a better effort on incorporating the complexities of the flow, turbulence–particle, particle–particle, and particle–wall interactions. However, to achieve this, it is mandatory to have access to efficient particle tracking algorithms able to carry out all these computations for millions of particles.

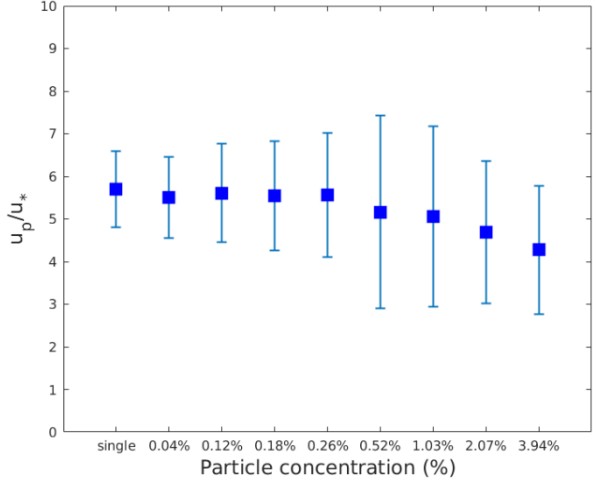
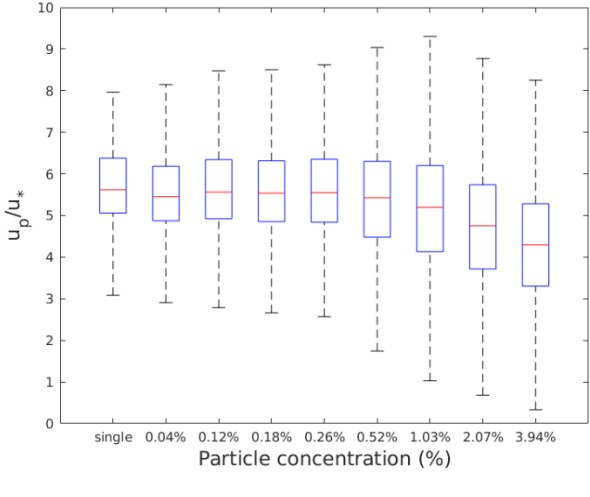

(**a**)                    (**b**)

**Figure 6.** Change of dimensionless particle velocity for a single particle and different particle concentrations. (**a**) Mean dimensionless particle velocity (blue square), where vertical blue lines represent two standard deviations. (**b**) Boxplot of dimensionless particle velocity; the red line depicts the median, whereas the box bottom and top blue lines indicate the 25th and 75th percentiles; vertical dash lines represent the most extreme values not considering outliers.

### 4.3. Computational Resources

The sub-models that form the proposed saltation model consist of the free flight of particles, the collision of particles with the bottom wall, and collisions among particles. Each one of them consumes computational resources that translate to computational time consumption. Sub-models may require more or less time depending on model considerations.

The analysis of the employed computational resources for running the simulations are quantified as computing time and presented in Figure 7. Four cases were investigated to address the proportion of time employed for calculating the velocity in a free-flight particle motion and the corresponding proportion of time employed for updating velocities after a collision with the bottom wall or after collision among particles. The four cases represent different concentrations. Case 1 includes 1000 particles and 0.04% concentration; case 2 considers 3000 particles and 0.12% concentration; case 3 corresponds to 6500 particles and 0.26% concentration, and, finally, case 4 does not include any type of collisions with 500 particles simulated.

Cases 1 to 3 include collisions among particles. For case 4, the collision sub-routine was turned off, to avoid particles colliding when they share the same position. For this latter case, the code assumes that particles overlap, without affecting their trajectories (there is no collision), as if they were ghost particles.

For the case where particles move freely, without collisions between them, 70% of computing time is employed to compute particle movement, whereas the remaining 30% is used to calculate particle velocities due to collisions with the wall.

For case 1, or 0.04% concentration, only 4.5% of the time is used to move particles while the remaining 95.5% is consumed to compute particle collisions (with the wall and between particles). The collision subroutine is activated whenever two particles collide, and velocities after collisions are modified based on the momentum exchange, thus modifying the particles trajectories. For case 2, 0.12% particle concentration, 98.4% of time is consumed on particle collisions. Finally, for case 3, 0.26% particle concentration, a modest 0.86% of time is used to move particles, while more than 99% of computing time is devoted to compute particle collisions.

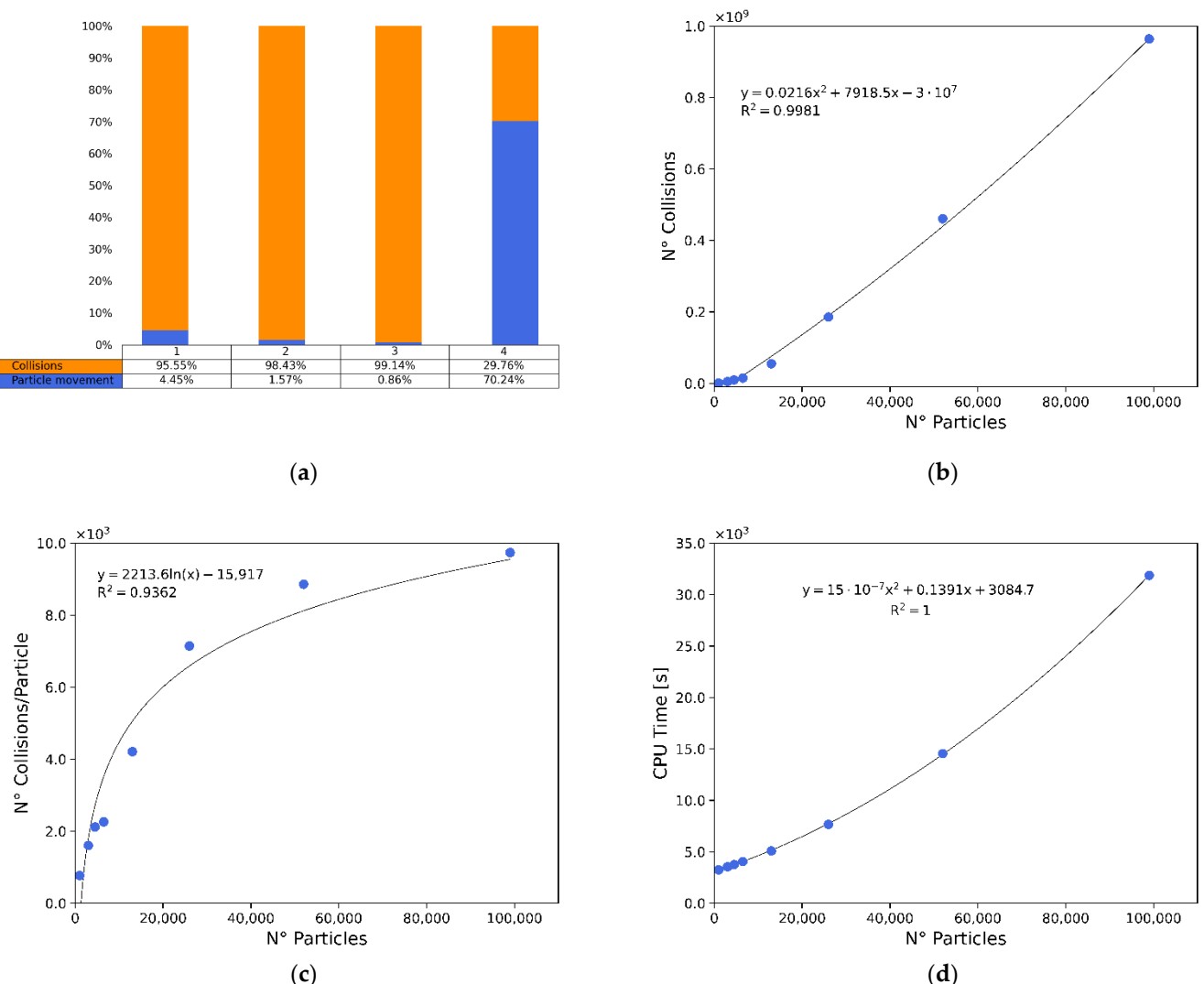

**Figure 7.** Computational time devoted to the simulation. (**a**) Comparison of computational time used calculating particle collisions (orange) and particle movement (blue), for 4 cases: 1000 (1), 3000 (2), 6500 (3), and 500 (4) particles, respectively, where in case 4 no interparticle collisions are allowed by the model; (**b**) number of interparticle collisions versus number of particles simulated; (**c**) number of interparticle collisions normalized by number of particles simulated; and (**d**) CPU time (in seconds) devoted to each multiparticle simulation.

The results reveal that, as concentration increases, more computational resources are required to calculate new velocities due to collisions. This difference is explained by the increase in the number of collisions during the simulation. In Figure 7b–d, the statistics of collisions among particles are presented. The second order polynomial from Figure 7b illustrates how the number of collisions grows when the concentration increases. In addition, Figure 7c depicts how the number of collisions experienced by each particle starts to reach a plateau as concentration increases. Every time there is a collision with another particle, the particle–particle collision subroutine is activated, increasing the computational costs. In addition to this, the increase in particles in neighboring cells also increases the computational efforts, as more particles need to be verified for potential collisions at each time step. This may become a major issue when increasing the computational domain and/or the number of particles in a given simulation. Figure 7d shows how CPU time presents a quadratic growth as particle concentration increases.

*4.4. Particle Diffusion*

The diffusion process of sediment particles moving as bedload can be mathematically described as the time evolution of the variance of the particle location in the streamwise (X) and spanwise (Z) directions [42,43]. Four (out of 24) examples of time evolution of the second order moment (variance) of particle location in the spanwise and streamwise directions at two different concentrations (0.12% and 3.94%) and the same flow intensity ($\tau_* / \tau_{*c} = 1.79$) are shown in Figure 8. No additional time evolutions are included for the sake of clarity. The blue and orange circles indicate values from the simulation, whereas black lines indicate the slope obtained by lineal regression. The exponents of $t$ right above each slope black line indicate the result of the $\gamma$ parameter. The time evolution of the particle location indicates how diffusive the movement of particles are in the local range (diffusion of particles between collisions with the wall) and in the intermediate range (diffusion of particles over all jumps, before particles come to rest). According to Nikora et al., it is expected that the local range diffusion exponents should be closer to one, i.e., $\gamma \sim 1$ (where $\sim t^{\gamma}$), indicating ballistic diffusion. However, for the intermediate range, the exponent should be smaller than 1, i.e., $\gamma < 1$. Values of $\gamma = 0.5$ indicate normal diffusion, whereas values of $\gamma > 0.5$ and $\gamma < 0.5$ indicate superdiffusion and subdiffusion, respectively. Figure 8a,b show the simulation results for extreme cases for the time evolution of the location of particles in the streamwise direction (X-direction) for 3000 and 99,000 particles. In Figure 8a,b, it is clear that local diffusion is slightly affected by the increase in concentration, as $\gamma$ changes from 1 to 0.9, denoting a slight reduction in local diffusion as concentrations are increased up to $\sim 5\%$. However, changes in diffusion at the intermediate rage are larger, as the values of $\gamma$ increase from 0.6 up to 0.85. The latter indicates that the increase in particle concentration from 0.12% to nearly 5% causes a great increase in particle diffusion, moving from nearly normal diffusion up to a solid superdiffusion. These results indicate that at the local range, changes in particle movement over time between collisions with the wall in the X-direction slightly restrict the particle movement, whereas at the intermediate range, particle movement increases importantly as particle concentrations are augmented. This can be explained by the increase in particle–particle collisions, which restricts the movement of particles locally, but at the same time causes a larger change in particle location at the intermediate range. A similar effect occurs when analyzing the particle movement in the transverse direction (Figure 8c,d). The increase in particle concentration limits the local movement of particles in Z while increasing the changes in particle location at the intermediate range, triggered by the increase in particle–particle collisions.

In Figure 9, the results for the diffusion analysis are shown for a total of 24 cases, which include eight particle concentrations and three degrees of flow intensity for each particle concentration. Figure 9a,b, correspond to diffusion in the X-direction for the local and intermediate ranges, respectively, whereas Figure 9c,d show the diffusion in the Z-direction for the local and intermediate ranges, respectively. Diffusion figures follow the conceptual model of diffusion for sedimentary particles proposed by [39,40]. Equations obtained for each range allow quantification of diffusion through the parameter $\gamma$. Tables 4–7 and Figure 9 summarize values of $\gamma_x$ and $\gamma_z$ for both the local and the intermediate range, according to the conceptual model of [39].

Results from Figure 9a show that diffusion in the streamwise direction is only affected (decreased) at the local range for particle concentrations of $\sim 5\%$ for lower shear stresses (or flow intensities). However, when the flow intensities increase, particle diffusion decreases for particle concentrations of 1% and above. Instead, for the intermediate range in the streamwise direction (Figure 9b), the increase in particle diffusion can be seen from low concentrations (0.52% and above). Again, all these changes can be explained by the increasing limitation of movement of particles at the local range as space is being populated by more particles. Yet, this also causes an intensification of the rate of collisions among particles as concentrations grow, causing an increase in changes of direction, which translates to a greater diffusivity. A similar analysis can be obtained from Figure 9c, where local diffusion

in the spanwise direction is decreased for all flow intensities when particle concentrations are larger than 2%. Instead, in the intermediate range, in the spanwise direction (Figure 9d), an increase in particle diffusion can be seen in all flow intensities, even in concentrations lower than 1%. The same reasoning stated for movement restrictions at the local range and an increase in the rate of direction changes triggered by particle–particle collisions can be argued, as previously explained.

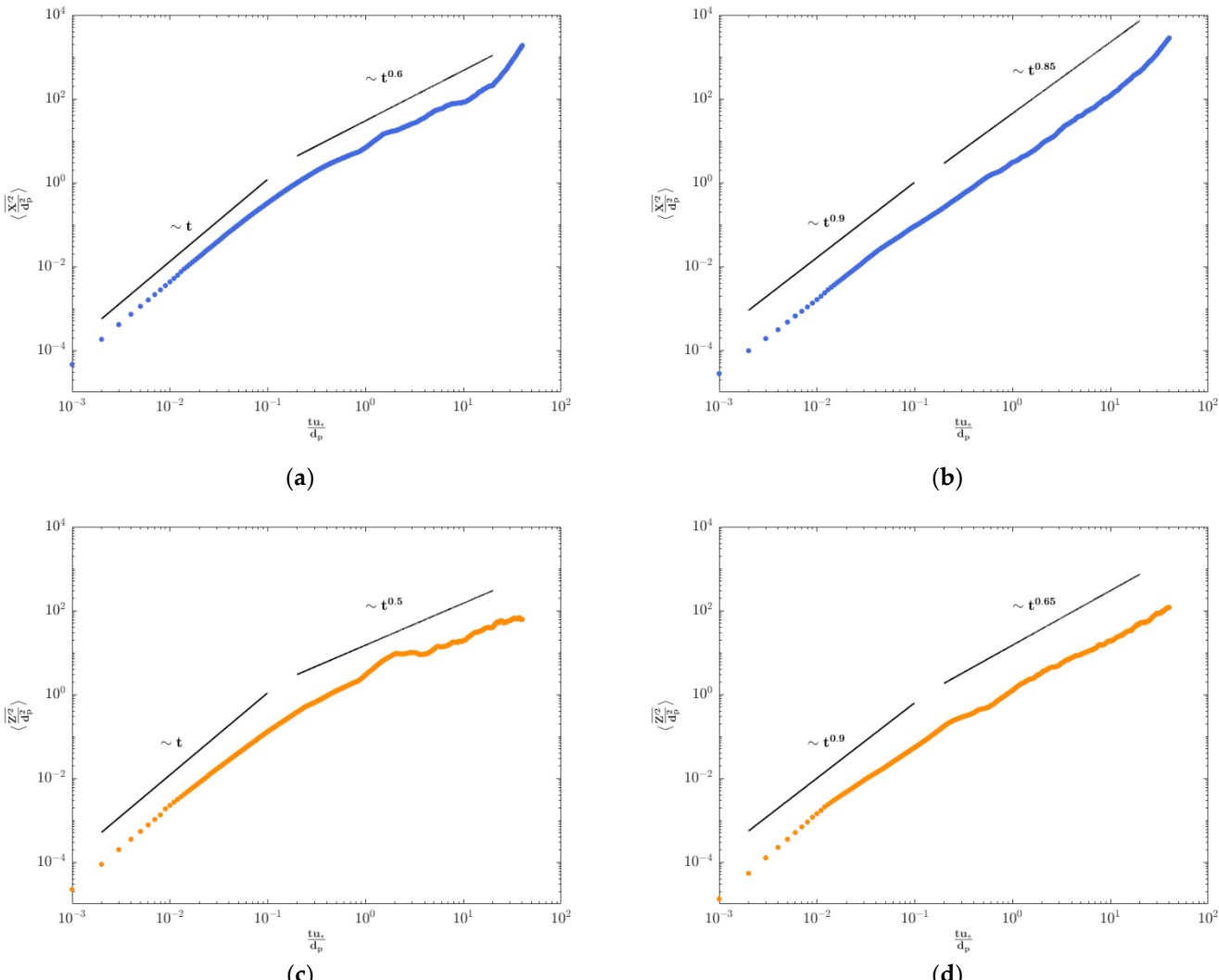

**Figure 8.** Dimensionless time evolution of the dimensionless particle location variance in the streamwise, $\langle \overline{X'^2}/d_p^2 \rangle$, and spanwise, $\langle \overline{Z'^2}/d_p^2 \rangle$, directions, following the conceptual method presented by Nikora et al. [39]: (**a**) 3000 particles (concentration of 0.12%); (**b**) 99,000 particles (concentration of 3.94%); (**c**) 3000 particles (concentration of 0.12%); and (**d**) 99,000 particles (concentration of 3.94%). Blue and red filled circles indicate simulation results, whereas solid black lines depict slopes for the local (left) and intermediate (right) ranges; $\tau_* / \tau_{*c} = 1.79$.

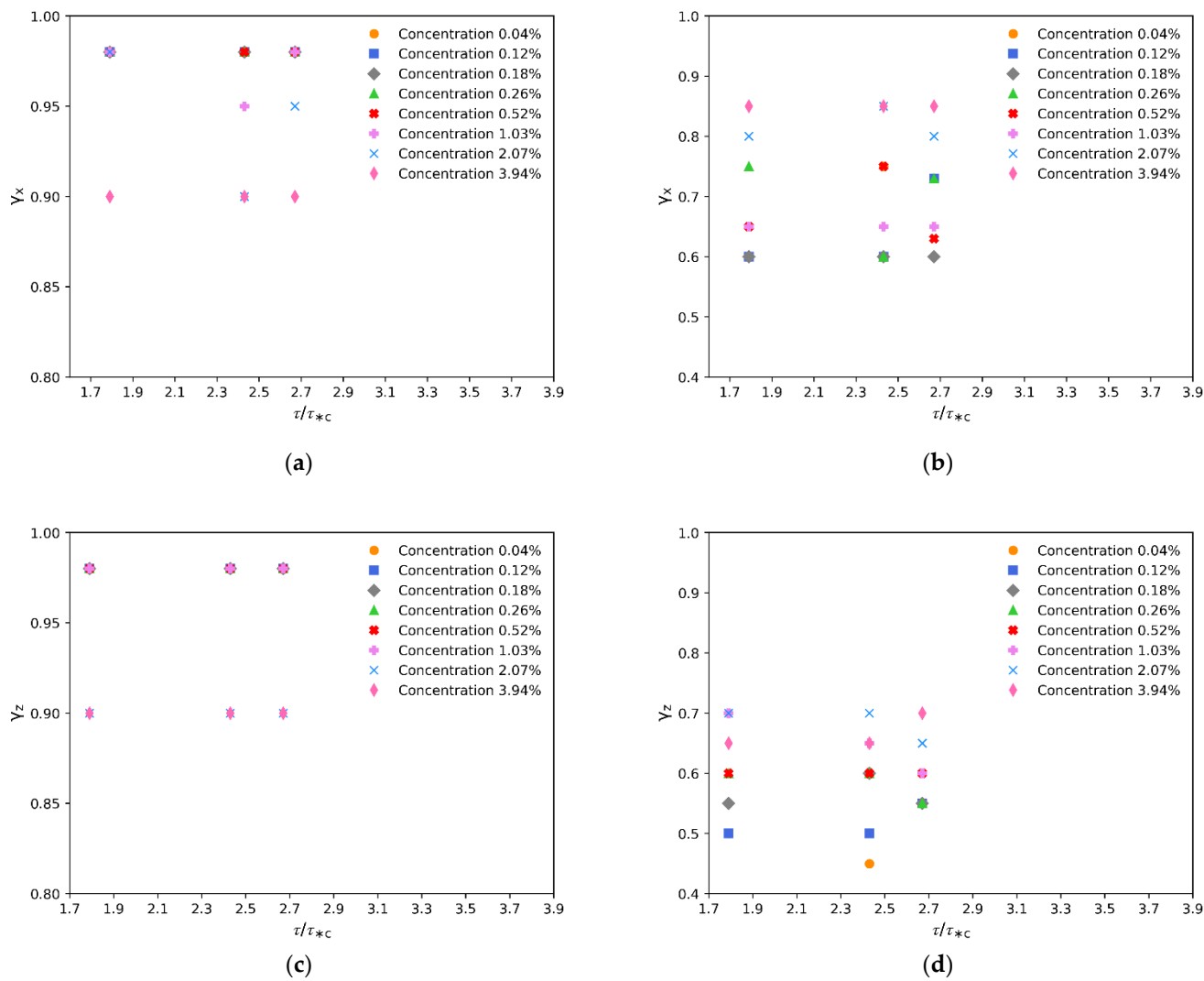

**Figure 9.** Diffusion results in the X and Z directions for local (ballistic) and intermediate ranges at different particle concentrations. Streamwise diffusion ($\gamma_x$) (**a**) for the ballistic and (**b**) the intermediate range. Spanwise diffusion ($\gamma_z$) (**c**) for the local and (**d**) the intermediate range.

**Table 4.** Summary of values for $\gamma_x$ for particle diffusion in the local range in the X-axis.

| Flow Intensity ($\tau_*/\tau_{*_c}$) | Local Range: $\gamma_x$ | | | | | | | |
|:---:|:---:|:---:|:---:|:---:|:---:|:---:|:---:|:---:|
| | Particle Concentration (%) | | | | | | | |
| | **0.04** | **0.12** | **0.18** | **0.26** | **0.52** | **1.03** | **2.07** | **3.94** |
| 1.79 | 0.98 | 0.98 | 0.98 | 0.98 | 0.98 | 0.98 | 0.98 | 0.90 |
| 2.43 | 0.98 | 0.98 | 0.98 | 0.98 | 0.98 | 0.95 | 0.90 | 0.90 |
| 2.67 | 0.98 | 0.98 | 0.98 | 0.98 | 0.98 | 0.98 | 0.95 | 0.90 |

**Table 5.** Summary of values for $\gamma_x$ for particle diffusion in the intermediate range in the X-axis.

| Flow Intensity ($\tau_*/\tau_{*c}$) | Intermediate Range: $\gamma_x$ | | | | | | | |
|---|---|---|---|---|---|---|---|---|
| | Particle Concentration (%) | | | | | | | |
| | 0.04 | 0.12 | 0.18 | 0.26 | 0.52 | 1.03 | 2.07 | 3.94 |
| 1.79 | 0.60 | 0.60 | 0.60 | 0.75 | 0.65 | 0.65 | 0.80 | 0.85 |
| 2.43 | 0.75 | 0.60 | 0.60 | 0.60 | 0.75 | 0.65 | 0.85 | 0.85 |
| 2.67 | 0.73 | 0.73 | 0.60 | 0.73 | 0.63 | 0.65 | 0.80 | 0.85 |

**Table 6.** Summary of values for $\gamma_z$ for particle diffusion in the local range in the Z-axis.

| Flow Intensity ($\tau_*/\tau_{*c}$) | Local Range: $\gamma_z$ | | | | | | | |
|---|---|---|---|---|---|---|---|---|
| | Particle Concentration (%) | | | | | | | |
| | 0.04 | 0.12 | 0.18 | 0.26 | 0.52 | 1.03 | 2.07 | 3.94 |
| 1.79 | 0.98 | 0.98 | 0.98 | 0.98 | 0.98 | 0.98 | 0.90 | 0.90 |
| 2.43 | 0.98 | 0.98 | 0.98 | 0.98 | 0.98 | 0.98 | 0.90 | 0.90 |
| 2.67 | 0.98 | 0.98 | 0.98 | 0.98 | 0.98 | 0.98 | 0.90 | 0.90 |

**Table 7.** Summary of values for $\gamma_z$ for particle diffusion in the intermediate range in the Z-axis.

| Flow Intensity ($\tau_*/\tau_{*c}$) | Intermediate Range: $\gamma_z$ | | | | | | | |
|---|---|---|---|---|---|---|---|---|
| | Particle Concentration (%) | | | | | | | |
| | 0.04 | 0.12 | 0.18 | 0.26 | 0.52 | 1.03 | 2.07 | 3.94 |
| 1.79 | 0.50 | 0.50 | 0.55 | 0.60 | 0.60 | 0.70 | 0.70 | 0.65 |
| 2.43 | 0.45 | 0.50 | 0.60 | 0.60 | 0.60 | 0.65 | 0.70 | 0.65 |
| 2.67 | 0.55 | 0.55 | 0.55 | 0.55 | 0.60 | 0.60 | 0.65 | 0.70 |

From the above results (Tables 4–7), it could be stated that, when analyzing the increase in the diffusion of particles at the intermediate range for the flow intensities of $\tau_*/\tau_{*c} = 2.43$, 2.67, the diffusion in the X-direction, $\gamma_x$, is 20% larger than the diffusion in the Z-direction ($\gamma_z$), whereas for the smallest flow intensity, $\tau_*/\tau_{*c} = 1.79$, the diffusion in the X-direction, $\gamma_x$, is only 15% larger than the diffusion in the Z-direction ($\gamma_z$), showing a slight effect of the flow intensities in the particle diffusion. The latter suggests that at higher flow intensities the flow counteracts the effect of the diffusion in the Z-direction in a value close to 5%. This makes sense, as the main flow direction is set in the X-direction, hence the changes in the transverse direction caused by the changes of momentum due to particle collisions are counteracted by the momentum transferred by the flow to the moving particles. In addition, when analyzing the effect of diffusion in the intermediate range, for a fixed flow intensity, there seems to be a trend indicating that, for larger concentrations, the particles tend to increase the changes in location due to the increase in particle–particle collisions, which translates to an increase in particle diffusion.

## 5. Conclusions

A C++ decoupled saltation code has been implemented and validated (for sands) for a wide range of particle concentrations in the dilute range, from 0.04% to 3.94%, and three different flow intensities, corresponding to non-dimensional shear stresses of $\tau_*/\tau_{*c} = 1.79$, 2.43, 2.67. The saltation code has been extended from a previous code and optimized in terms of computational efforts, making possible the simulation of close to 100,000 particles, while improving the efficiency of the particle–particle collision subroutine and taking advantage of multiprocessor computing. The following conclusions can be extracted from the simulations:

1.　The particle model results, coupled with a high-resolution LES-WALE model, are in good agreement with the experimental information available in the literature for

the saltation motion of sands. The particle-jump statistics follow the mean trend and variance of experimental results.

2. In the local range and at concentrations smaller than 2%, both diffusion coefficients $\gamma_x$ and $\gamma_z$ are equal to 0.98. This value is very close to the theoretical value of 1.0 proposed by Nikora et al. for ballistic motion. At the largest concentration of 3.94%, both coefficients decrease to a value of 0.9 regardless of the flow intensity. This slightly smaller value is attributed to the effect of the increase in particle concentration, which restricts the particle movement al the local scale.

3. In the intermediate range, it is very clear that $\gamma_x$ greatly increases, and it is on average larger than $\gamma_z$, which also increases. For $\tau_*/\tau_{*c} = 2.43$, 2.67, $\gamma_x \sim 1.2\gamma_z$, whereas for the smallest flow intensity, $\tau_*/\tau_{*c} = 1.79$, $\gamma_x \sim 1.15\gamma_z$. This suggests that the magnitude of the changes of particle direction in the streamwise direction caused by the momentum exchange of particle collisions are counteracted by the increase in the momentum exchange from the flow to the moving particles as the flow intensity rises.

4. For a fixed flow intensity, there is a trend in the intermediate range to have larger diffusion coefficients as particle concentration increases. This is caused by the increase in the interparticle collision rates.

5. When the number of particles is greater than 13,000 it was observed that the number of collisions grows linearly with the number of particles. However, the number of collisions per particle reaches a plateau, which is an indication that there exists an upper limiting value for the number of collisions per particle.

6. Larger particle concentrations cause a reduction in mean particle jump height and length for lower flow intensities. As flow intensities increase, this effect still occurs but with a smaller magnitude.

7. The computational effort of the model devoted to particle collisions becomes limiting as the particle concentrations increase within the dilute flow range. This is a very important aspect to account for when increasing the computational domain (larger amounts of particles within the dilute range) for real scale engineering structures, or when trying to mimic non-dilute conditions (concentrations higher than 5%, where two-way coupling must be also considered).

8. The effect of particle concentration on particle velocity is important in two aspects: First, a significant reduction on its mean magnitude is computed in flows with sediment concentrations above 0.52%. Second, there is an increase in the variance of particle velocities for higher particle concentrations. More importantly, given that one particle may move nine times faster than other particles (under the same flow intensity for a given particle concentration), it thus introduces larger fluctuations in the bedload transport rates when compared to mean values. This is a very important finding, as most of the equations used to estimate sediment transport rates use mean values, not considering the fluctuations in particle statistics.

9. Particle velocity is an important variable when computing sediment transport rates. If a small computational domain, limited simulation time, assuming particles of the same diameter, with fixed particles at the bed, moderate Reynolds numbers, with no changes of geometry or lateral flows (due to stream spatial changes) generates large particle velocity fluctuations, one can only imagine how the fluctuations will (most likely) increase up to a point where differences between sediment transport estimations and measurements can differ by orders of magnitude.

10. Particle tracking codes such as the one used for this study may help understand the complex interactions occurring at the microscale (flow–particle, particle–particle, and particle–wall interactions) to find new expressions for the computation of rates of sediment transport that take into account its fluctuations.

**Author Contributions:** Conceptualization: P.A.M.-C., J.P.T., J.P., E.G. and J.A.A.; Methodology: P.A.M.-C., J.P.T., J.P. and J.A.A.; Software: P.A.M.-C., E.G., J.P. and J.A.A.; Validation: P.A.M.-C., E.G. and S.S.; Formal analysis: P.A.M.-C., J.P.T., E.G. and S.S.; Investigation: P.A.M.-C., E.G. and S.S.;

Resourses: P.A.M.-C. and J.P.; Writing original draft preparation: P.A.M.-C., J.P.T., E.G., S.S. and J.P.; Writing original draft revision: P.A.M.-C., J.P.T., E.G., S.S., J.P. and J.A.A.; Supervision: P.A.M.-C., J.P.T. and J.A.A.; Project administration: P.A.M.-C.; Funding Acquisition: P.A.M.-C. and S.S. All authors have read and agreed to the published version of the manuscript.

**Funding:** This research was funded by FAI Puente Regular grant number INV-PR-2022-01.

**Institutional Review Board Statement:** Not applicable.

**Informed Consent Statement:** Not applicable.

**Data Availability Statement:** The data presented in this study is publicly available in the cited papers, and the average and standard values of the simulated outputs are contained within the article.

**Conflicts of Interest:** The authors declare no conflict of interest.

## Abbreviations

| | |
|---|---|
| $q^*$ | Dimensionless bedload transport rate |
| $\tau^*$ | Shields parameter |
| $\tau_c^*$ | Critical Shields parameter |
| $R$ | Submerged specific gravity of the particle |
| $m$ | Particle mass |
| $u_p$, $v_p$, $w_p$ | Particle velocity in the X, Y and Z directions |
| $u_f$, $v_f$, $w_f$ | Flow velocity in the X, Y and Z directions |
| $\boldsymbol{F_{sw}}$ | Submerged weight |
| $\boldsymbol{F_{dr}}$ | Drag force |
| $\boldsymbol{F_{lf}}$ | Lift force |
| $\boldsymbol{F_{bs}}$ | Basset force |
| $\boldsymbol{F_{mg}}$ | Magnus force |
| $\boldsymbol{F_{am}}$ | Added mass force |
| $\boldsymbol{F_{fa}}$ | Fluid acceleration force |
| $\rho$ | Water density |
| $\rho_p$ | Particle density |
| $g$ | Acceleration of gravity |
| $d_p$ | Particle diameter |
| $|\boldsymbol{u_r}|$ | Magnitude of the relative particle velocity |
| $C_D$ | Drag coefficient |
| $A$ | Particle cross section in the direction of $u_r$ |
| $U_r$ | Magnitude of $u_r$ |
| $C_L$ | Lift coefficient |
| $|\boldsymbol{u_r}|_T$ | Particle relative velocity at the top region |
| $|\boldsymbol{u_r}|_B$ | Particle relative velocity at the bottom region |
| $t$ | Time |
| $\mu$ | Dynamic fluid viscosity |
| $\tau$ | Shear stress |
| $\boldsymbol{\Omega_p}$ | Angular velocity of the particle |
| $C_M$ | Magnus coefficient |
| $C_m$ | Added mass coefficient |
| $u_*$ | Shear velocity |
| $\alpha$ | Factorization coefficient |
| $R_p$ | Explicit Reynolds number of the particle |
| $\nu$ | Kinematic fluid viscosity |
| $\theta$ | Angle formed by the bed and the horizontal plane |
| $\varpi_y$ | Non-dimensional component of the rotation vector in the spanwise direction |
| $\omega_y$ | Angular velocity of the particle along the spanwise direction |

| | |
|---|---|
| $y$ | Direction normal to the channel bed |
| $d(\cdot)/dt$ | Material derivate |
| $Re_p$ | Particle Reynolds number |
| $v_s$ | Particle fall velocity |
| $\boldsymbol{\varpi}$ | Non-dimensional particle rotation vector |
| $C_t$ | Coefficient used in the computation of the particle angular momentum equation |
| $C_1$, $C_2$, $C_3$ | Coefficients used in the computation of the particle angular momentum equation |
| $\boldsymbol{\varpi_r}$ | Non-dimensional vector of relative rotation of the particle with respect to the fluid vorticity |
| $e$ | Restitution coefficient |
| $f$ | Friction coefficient |
| $\gamma_x$ | Diffusion factor on the X-axis |
| $\gamma_z$ | Diffusion factor on the Z-axis |
| $X'^2$, $Z'^2$ | Second order moments of the particle position in X and Z |
| $\alpha_x$, $\alpha_z$ | Constant coefficient used in the computation of the particle diffusion in the X- and Z-directions |
| $X'$, $Z'$ | First order moment of the particle position in X and Z |
| $\langle\ \rangle$ | Ensemble average |
| $H$ | Particle jump height |
| $L$ | Particle jump length |
| $h$ | Channel height |
| $Re_\tau$ | Friction Reynolds number |
| $Re$ | Reynolds number |
| $\overline{y_1^+}$ | Y-plus mean value measured from the wall to the first finite volume |
| $y_1$ | Distance from the wall to half of the first finite volume |

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
