# Peer review of "The Effect of Particle Concentration on Bed Particle Diffusion in Dilute Flows"

_water, doi:10.3390/w14193105_

Round 1

Reviewer 1 Report

Water-1899745v1

This is an interesting numerical work of the effect of particle concentration on bed particle diffusion with useful applications to sediment transport and deposition. The text reads well with only minor language errors and a few technical deficiencies. The authors should avoid detailed description on known items. Some suggestions for improvement are as follows:

01. Line 17: Improve the clarity of the sentence “The model is one-way coupled to a validated turbulent LES channel flow”.

02. Line 20: Define or explain what is “flow intensities”. Do the authors mean flow velocity or Reynolds number?

03. Page 5: In Figure 1, the z-axis of the coordinate system should be in the vertical direction as mentioned in the text.

04. L411: Better to use as heading “4. Results and discussion”.

05. L499 & 543: There are to sections headed “4.3 Computational resources”.

06. L221-222: List references correctly, complete, consistently and according to the Journal standards.

07. L632-633: Complete the listing of reference [2].

08. L634-635: Complete the listing of reference [3].

09. L640: Complete the listing of reference [6].

10. L645-646: Correct the listing of reference [9].

11. L668-669: Complete the listing of reference [20].

12. L698: In reference [35], write “Dept. of Civil Eng.” instead of “D. of C. E., Ed.”

Reviewer 2 Report

The general impression is that the paper covers a very interesting modeling approach of sediment motion. The main issue is that it remains unclear what is the novelty in the presented paper. If I understand correctly, the presented numerical model is an existing and published model, and the results are compared to existing and published measurements. 

If this is not the case I would urge the Authors to include significant alterations in the presentation of their contribution within the overall Manuscript and to make sure they clearly explain the elements of the research that are the result of their work from those elements they simply copied from others. At the present form of the Paper it is impossible to conduct a proper review as there is too much key information missing.

Some more specific issues are listed below.

It is not clear if the Authors simply employed an existing modeling approach to create a numerical code for their own needs, or if they implemented certain changes to an existing numerical approach. Could the Authors please clarify this. If they included certain changes to the model, than it is acceptable to simply cite the resource they used for the essential model described in Section 2. In that case it should be clearly outlined what exactly is their input, that differs when compared to the cited resource. 

Otherwise, if they developed a model that is different from other existing models, Section 2. should be extended to include more explanations regarding the derivation of the listed equations.

Regarding Eqs. 9, 10 and 11, please provide additional information regarding their origin, derivation, employment, constraints, implemented assumptions and other information required to comprehend the provided information.

Lines 310 and 311, sentences similar to this "The computational simulation 310 follows the guidelines presented in another study [45]" should be avoided as they are useless. They provide no information.

The Authors should provide a more specific description of the utilized measurements.

Reviewer 3 Report

In the presented paper entitled "The effect of particle concentration on bed particle diffusion in dilute flows" the simulation results of a Lagrangian particle tracking model with a 3D sub-model proposed by Authors are discussed. It is written in very good English. The introduction part extensively describes the theory behind the models used and justifies the choices made to tackle the problem. Unfortunately, the description of the obtained results has to be taken care of before further processing of this paper. All comments are presented below.

1. As there is a large number of symbols used I would recommend to give a list of symbols.

2. Figure 1 should give the idea of particle jumps. I would suggest to describe it a bit in the main text.

3. Figure 2: either it is unnecessary or it should be described in text. There is no mention anywhere about it.

4. Figure 3: The grid distribution part (Fig. 3a) does not give any valuable information. In fact, the description also fails. Is there and boundary layer? How is it built? 

5. Section 3.2: What are other conditions of simulation? What software was used? 

6. L354: Reference [64] does not exist in the bibliography of this paper.

7. L404-405: Information repeated "number of collisions among particles is also considered"

8. Figure 5: The standard deviation marked in the graphs are very large both in simulation and in experimental cases. Is the situation the same for rough data? Is there any way to diminish them, at least in the case of simulation? In the case of experiments I understand that (i) the results are from literature and  (ii) sometimes the errors can be large despite every precaution taken. But simulations are to make the best possible result and in the presented case is seems that there was not enough care given to reproducibility of the results.

9. Section 4.2, description of Figure 6: The description is from some other research. The values that are used (consistently) are not even close to the range of the simulations. It is as in this place a part of some other article was included. 

10. Figure 7b: The description in text is insufficient. Please elaborate more on that.

11. Figure 9: The same problem as in the case of Fig.7b. Something is presented, but I get the impression that either Authors do not know how to describe it or they do not want to describe it for some reason. Please tell us more.

Also the caption to this figure is incomplete - please make it clear which graph presents which parameter.

12. In the case of describing the Figures presented in the articles, Authors should consider what they want to present and discuss accordingly. In this version of the manuscript the presentation of the results is chaotic and incomplete. Moreover, some of the conclusions are obvious, some are out of nowhere (no such things were even suggested in text) and there is no conclusion telling the reader if the applied models properly reflects the experimental results or not.

13. There are a few spelling mistakes - please check lines 116, 168, table 1, 489, 551.

Round 2

Reviewer 2 Report

Thank you for the implemented corrections.

Reviewer 3 Report

Thank you for your responses. I accept your answers.